# CTRLSYNTH: CONTROLLABLE IMAGE TEXT SYNTHESIS FOR DATA-EFFICIENT MULTIMODAL LEARNING

## ABSTRACT

Pretraining robust vision or multimodal foundation models (*e.g.*, CLIP) relies on large-scale datasets that may be noisy, potentially misaligned, and have long-tail distributions. Previous works have shown promising results in augmenting datasets by generating synthetic samples. However, they only support domain-specific ad hoc use cases (*e.g.*, either image or text only, but not both), and are limited in data diversity due to a lack of fine-grained control over the synthesis process. In this paper, we design a *controllable* image-text synthesis pipeline, CtrlSynth, for data-efficient and robust multimodal learning. The key idea is to decompose the visual semantics of an image into basic elements, apply user-specified control policies (*e.g.*, remove, add, or replace operations), and recompose them to synthesize images or texts. The decompose and recompose feature in CtrlSynth allows users to control data synthesis in a fine-grained manner by defining customized control policies to manipulate the basic elements. CtrlSynth leverages the capabilities of pretrained foundation models such as large language models or diffusion models to reason and recompose basic elements such that synthetic samples are natural and composed in diverse ways. CtrlSynth is a closed-loop, training-free, and modular framework, making it easy to support different pretrained models. With extensive experiments on 31 datasets spanning different vision and vision-language tasks, we show that CtrlSynth substantially improves zero-shot classification, image-text retrieval, and compositional reasoning performance of CLIP models.

## 1 INTRODUCTION

High-quality large-scale datasets have driven the success of large foundational AI models (Radford et al., 2021; Rombach et al., 2022; Touvron et al., 2023). Collecting and annotating datasets at large-scale is challenging and costly. One solution is to crawl data from the web; however, web data is noisy (Lai et al., 2024; Kang et al., 2023), has long-tail distributions (Udandarao et al., 2024), and often causes privacy or copyright issues (Schuhmann et al., 2022). Synthetic data presents a viable and complementary alternative to overcome these challenges, as it allows for precise control over data generation and customization to meet specific requirements. A large body of work has focused on improving the quality of synthetic data for image and text data, from the generation of high-quality images (Dunlap et al., 2023; Islam et al., 2024) to the improvement of synthetic captions (Lai et al., 2024; Fan et al., 2023). While these works have shown that synthetic data successfully improves model performance for various vision or vision-language tasks, their synthetic pipeline is often ad hoc and tailored to specific purposes such as training better CLIP models or improving domain-specific vision models (*e.g.*, DiffuseMix uses diffusion models to augment images and improves accuracy on image classification tasks Islam et al., 2024). These data synthesis works also lack explicit fine-grained control over the generated texts or images, which are important for tasks with long-tail distribution (*e.g.*, augmenting tail class samples) or enforcing safety requirements (*e.g.*, mitigating biased or sensitive content generation Schramowski et al., 2023).

In this work, we aim to systematically control the synthetic pipeline for generating image-text data while accommodating different use cases (*e.g.*, improving long-tail task performance, enhancing compositional reasoning of CLIP models, etc.). Our intuition is that large foundation models are already pretrained on a wide range of data and contain general knowledge about concepts, objects, and their relationships. For example, text-to-image models (*e.g.*, Rombach et al., 2022; Podell et al., 2024) can generate detailed high-quality images based on text instructions. Similarly, large language models

Figure 1: CtrlSynth: A modular, closed-loop, controllable data synthesis system. The *oval nodes* indicate that the pretrained models and *rounded boxes* represent text or image data. The text and image controllers are used to guide the data synthesis.

(LLMs) (*e.g.*, OpenAI, 2022; Touvron et al., 2023) have strong instruction-following capabilities, which can be used to control the text data generation. CtrlSynth leverages these large pretrained models to build a modular and controllable synthetic data generation pipeline. CtrlSynth allows users to apply explicit control instructions to guide data generation for images and texts. Unlike previous data synthesis works that use image-captioning models to directly generate captions given an image (*e.g.*, Li et al., 2024; Lai et al., 2024), CtrlSynth decomposes image-to-text generation process into two separate steps, providing more fine-grained control to users for synthesizing data. Figure 1 shows an overall architecture of the CtrlSynth pipeline. For an input image, CtrlSynth first uses a pretrained vision model to extract key objects, attributes, and their relations as visual tags. It then uses a text controller to create text synthesis instructions and guide the LLM to use visual tags to generate high-quality text outputs. Similarly, we devise an image controller that steers how the text prompts (or caption) can be used to guide the diffusion model to generate a desired image. Users can also feed the generated synthetic images into the tagging model again, forming a closed-loop data pipeline. Then users can start with synthetic or original images and texts and further generate more image-text pairs. The text and image controllers are modular, allowing users to control any part of the text or image generation process.

Compared to previous works, CtrlSynth provides three main benefits: (1) **Controllable synthesis**: CtrlSynth allows users to define policies on the visual tags or texts; enabling granular control over text and image generation; (2) **Closed-loop system**: CtrlSynth requires no additional training and can synthesize text from images and vice-versa using existing pretrained models. This closed-loop design additionally provides automatic filtering and verification capabilities to discard undesirable synthetic samples without manual or heuristics-based rules. (3) **Flexible and scalable**: CtrlSynth is modular and allows users to change its components (*e.g.*, pretrained models) easily. We evaluate the effectiveness of CtrlSynth on different tasks (*e.g.*, image classification, image-text retrieval, compositional reasoning, and long-tail tasks), covering **31 datasets** for vision and vision-language domains. We observe that CtrlSynth generated data improves the accuracy by (a) 23.4% on retrieval tasks, (b) 5% on the SugarCrepe compositional reasoning benchmark, and (c) $16\% \sim 21\%$ for long-tail vision tasks.

## 2 RELATED WORK

**Data-Efficient Vision-Language Representation Learning.** Contrastive Language-Image Pretraining (CLIP) (Radford et al., 2021) has popularized visual representation learning from image-text pairs due to its strong zero-shot transfer capabilities. Many recent works have focused on improving the data efficiency of training CLIP models. SLIP (Mu et al., 2022) brings self-supervised learning into a multitask learning framework to improve CLIP performance. FLIP (Li et al., 2023c) masks out image patches during CLIP training, improving training efficiency and zero-shot accuracy over baselines. CLIPA (Li et al., 2023b;a) further improves over FLIP ideas and reduces the number of image text tokens by block and syntax masking for CLIP training and it significantly reduces the training costs of CLIP models. LiT (Zhai et al., 2022) freezes the image encoder in CLIP models and achieves strong zero-shot transfer for CLIP models using much fewer data samples. All these techniques focus

on improving the training methods for CLIP models to enable better vision-language representations. CtrlSynth improves data augmentation for CLIP training by synthesizing diverse image text samples. Our method is orthogonal and could potentially benefit from these methods.

**Image-text Data Augmentation.** Much recent work aims to improve the caption quality of image-text pairs. For example, VeCLIP (Lai et al., 2024), LaCLIP (Fan et al., 2023), and ReCap (Li et al., 2024) leverage LLMs to synthesize new captions that are more informative and contain rich descriptions about the image. The key difference of CtrlSynth is that we provide more diverse and high-quality captions that outperform prior works (we will show in Table 5 and Table 6). This is because CtrlSynth breaks down the image semantics to allow more fine-grained control and recombination using LLM. Another line of work uses text-to-image models like diffusion models to generate synthetic images and augment downstream vision tasks. ALIA (Dunlap et al., 2023) uses language to guide the image editing process and provides domain-specific diversity to augment the image samples. DiffuseMix (Islam et al., 2024) augments image samples using diffusion models to blend original and generated images. EDA (Trabucco et al., 2023) generates variations of real images using diffusion models to maintain the semantics while augmenting image samples. These semantic image augmentation methods provide strong performance improvements on various vision datasets. Our CtrlSynth instead unifies the image and text synthesis via a closed-loop pipeline, it provides more flexibility and diverse synthetic samples while allowing more fine-grained control over the sample generation process. Prior image editing works like InstructPix2Pix (Brooks et al., 2023) and MagicBrush (Zhang et al., 2023) provide methods and datasets to enable precise control over image generation. While the image synthesis path in our pipeline could benefit from these works, our focus is to enable diverse data synthesis. It is an open research question to automatically generate the image editing instruction for each sample in a dataset. Our pipeline can also be combined with previous work (Mishra et al., 2024) to improve the performance of cross-domain retrieval tasks or when the target task has little real data to retrieve (Geng et al., 2024).

## 3 CTRLSYNTH

CtrlSynth leverages semantic knowledge and reasoning skills of pretrained foundation models (*e.g.*, large language and diffusion models) to generate diverse synthetic data samples in a controlled manner. Specifically, CtrlSynth consists of three foundation models: (1) a vision tagging model, (2) a large language model, and (3) a text-to-image model; plus the two text and image controllers. For a given real (①a in Figure 1) or synthetic (①c) input image, a *vision tagging model* (②a) extracts visual tags (*e.g.*, objects, attributes, and their relationships) (①e). These tags describe the image's visual concepts and semantic contexts. The *text controller* (③a) takes the image tags and user-defined control policies as inputs and generates instructions for synthesizing new text. An example control policy is to edit the tags or optionally add the text (①b) associated with the image. A *large language model* (②b) then follows the instructions and generates the synthetic text (①d). The *image controller* (③b) operates on the given input text and applies user-defined image control policies to output instructions for image synthesis. An example policy is to specify the style for generating artistic, cinematic, or realistic images. A *text-to-image model* (②c) takes an image synthesis instruction provided by the image controller as an input and produces a synthetic image as an output (①c).

### 3.1 KEY COMPONENTS

**Vision Tagging Model.** The goal of a vision tagging model (VTM) is to extract the basic visual elements (or tags) of an image, including all objects or entities, attributes (*e.g.*, color, shape, and size), and visual relations (*e.g.*, interaction between objects).

An example of extracting visual tags from VTM is shown in Figure 2. The tagging model can be either a multi-label image classifier (Mehta et al., 2024b) that predicts diverse tags in the image, or a black box system (*e.g.* an API service) that takes the input image and outputs the tags.

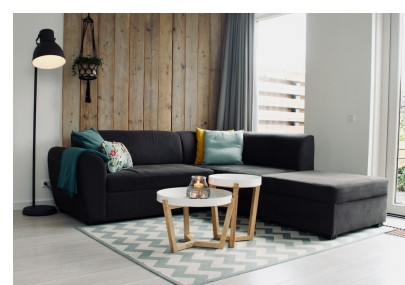

**Objects and attributes**: light candle, patterned rug, white coffee table, sectional sofa
**Relations**: in front of, on top, covered with

Figure 2: Visual tags of an example image[1]. Tags are non-exhaustive.

---

[1]Image credit: https://unsplash.com/photos/light-candle-on-round-white-coffee-table-and-sectional-sofa-GZ5cKOgeIB0

*Write a faithful caption by integrating the given phrases with the original sentence. Ensure any objects from the original caption are preserved while elaborating on the visual relationships and attributes provided in the phrases to create a more detailed depiction. Given sentence: {caption}. Given phrases: {phrases}. The caption should not contain any NSFW words. It should be grammatically correct. It should be concise, but not too short. Directly output the caption and do not add any formatting.*

Figure 3: An example instruction for LLMs to synthesize texts.

VTM, as a key component in CtrlSynth, can be a combination of an advanced captioning model (Xiao et al., 2024) that generates comprehensive image descriptions and an LLM that extracts the visual tags from the captions to decompose the visual semantics of an image into a set of fine-grained visual concepts. Appendix A.4 includes more details about this hybrid VTM. These fine-grained visual concepts can be easily modified and recomposed to create new visual contexts. This decompose-recompose feature of vision tags provides a large control space for synthesizing diverse texts.

Existing caption rewriting works (*e.g.*, VeCLIP (Lai et al., 2024)) rely on a multimodal captioning model to generate captions that are short sentences containing visual concepts. Image captions can be very descriptive but often only cover the most salient object of the scene, they are coarse-grained in structure (whole sentence or paragraph), and are hard to modify. Our key distinction is that VTM produces a comprehensive list of metadata information that describes the visual concepts in an image as completely as possible.

**Language Model.** Large language models (LLMs) have exhibited strong instruction-following capabilities. The goal of an LLM in CtrlSynth is to take an input textual instruction on how to generate a synthetic text that meets the requirements specified in the instruction. CtrlSynth employs the reasoning and composition capability of LLMs to recombine the visual image tags in the task instruction and compose new synthetic texts. The instruction for an LLM consists of three parts (Figure 3): *(i) task template* that specifies the details of the text synthesis task, *(ii) task content* that contains the actual visual tags (phrases) and an optional caption paired with the image, and *(iii) task constraint* that describes the style and formatting of the output text. Users can also apply custom policies over the instructions to guide the text synthesis process.

**Text-to-Image Model.** Text-to-image models generate novel and diverse image samples based on different input text prompts. CtrlSynth applies an image controller to account for the user-specified control policies and accordingly, updates the input text instructions from the previous step (i.e., language model). These updated instructions are then fed to text-to-image models for generating the image as an output. In our experiments, we use StableDiffusion models for text-to-image generation.

**Text and Image Controllers.** The controller in CtrlSynth is a function that takes an input text and transforms it into a specific text instruction for the LLM or text-to-image model.

The text controller accepts the visual tags of an image and a user-defined policy along with an optional original text as input and produces instructions to control the generation of synthetic text. In CtrlSynth, we study three predefined policies: (a) editing (remove, add, or replace) visual tags, (b) constraining the semantic meaning of a given sentence, and (c) styling the output text. Editing visual tags allows fine-grained control of synthetic visual content, for example, one can remove unwanted objects or attributes so they do not appear in the generated text. Constraining the meaning of synthetic text is useful in generating high-quality captions because many web-crawled captions are noisy. Enforcing the styling of output texts such as outputting into structured formats (*e.g.*, JSON) makes the texts easier to use in downstream tasks. In our experiments, we use 10 example text control policies for synthesizing image captions (see Appendix A.1 for details).

The image controller is similar to the text controller in functionality. It mainly steers image generation via specific prompting. We study two simple control policies to show the controllability and utility of CtrlSynth. The first one involves weighting particular tags in the input prompt (lower or increase individual weights for a given tag) so that the output image has a different focus on the objects or attributes. The second policy applies different styles (*e.g.*, cinematic, realistic, or art) to the output images for generating diverse content. Note that the control policies are flexible and can be easily modified for diverse use cases. For example, one can integrate more complex policies such as layout-guided (Lian et al., 2023) or planning-based (Yang et al., 2024b) image generation.

## 3.2 Image Text Synthesis in CtrlSynth

CtrlSynth is a modular and closed-loop system by design and supports diverse image and text synthesis configurations. In this section, we first introduce different synthesis paths in CtrlSynth and then describe how the closed-loop feature allows CtrlSynth to filter out low-quality samples.

**Flexible and diverse synthesis paths.** A data synthesis path ($SP$) starts and ends with a data node (rounded box in Figure 1). We define the following synthesis paths:

$SP(1)$: $1a \rightarrow 2a \rightarrow 1e \rightarrow 3a \rightarrow 2b \rightarrow 1d$. This path (Figure 4a) means CtrlSynth generates a new text that describes the original image. The synthetic text $1d$ may not align with the semantics in the original image since the LLM can create new combinations of the visual tags and add information that does not exist in the image. Such new information provides useful semantic augmentation over the original image while containing similar visual concepts.

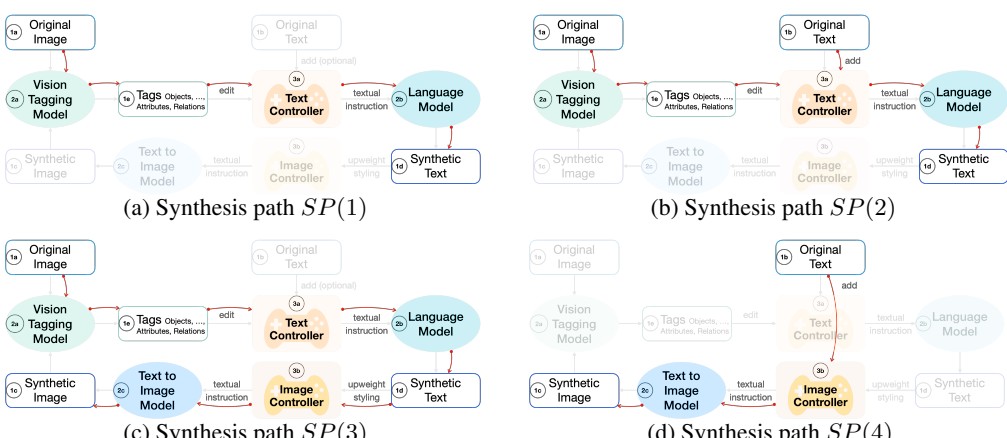

(a) Synthesis path $SP(1)$       (b) Synthesis path $SP(2)$

(c) Synthesis path $SP(3)$       (d) Synthesis path $SP(4)$

Figure 4: Different synthesis paths in CtrlSynth.

$SP(2)$: $1a \rightarrow 2a \rightarrow 1e \xrightarrow{1b} 3a \rightarrow 2b \rightarrow 1d$. This path (Figure 4b) is similar to the previous path but a key difference is that it constrains the synthetic text to be faithful[2] to an original text. We can consider it as using the VTM and LLM to synthesize an improved text over the original one. We will show later in Section 4.5 that text samples generated from this path outperform previous works (Lai et al., 2024; Fan et al., 2023) that rewrite noisy captions. We include the example prompts to reflect the control policies in Appendix A.1.

$SP(3)$: $1a \rightarrow 2a \rightarrow 1e \rightarrow 3a \rightarrow 2b \rightarrow 1d \rightarrow 3b \rightarrow 2c \rightarrow 1c$. This path (Figure 4c) provides both synthetic text ($1d$) and image ($1c$) samples. $1c$ can be an effective image sample that augments the original image ($1a$) or can be paired with ($1d$) to augment the original image-text pair ($1a$ and $1b$).

$SP(4)$: $1b \rightarrow 3b \rightarrow 2c \rightarrow 1c$. This path (Figure 4d) bypasses the language model and the original text is directly fed to the image controller and then generates a synthetic image ($1c$). The image sample could be a strong augmentation sample to the original image if the original text has a comprehensive and high-quality description.

Note that CtrlSynth supports more synthesis paths that are not listed above. For example, one can start with original text and use LLM to add creative elements and generate synthetic text and further use it to generate an image, i.e. $1b \rightarrow 3a \rightarrow 2b \rightarrow 1d \rightarrow 3b \rightarrow 2c \rightarrow 1c$. Another category of examples includes starting with synthetic texts or images and creating more synthetic samples.

**Self-filtering for better synthetic data.** Synthetic samples often suffer from degraded quality especially when running at large scale. Synthetic systems often rely on heuristics or rule-based filtering techniques to filter out bad-quality samples. Because CtrlSynth pipeline is closed-loop, it implicitly provides self-filtering functionality. To check the quality of the synthetic text, we detect if the synthetic text ($1d$) contains the visual tags ($1e$), to filter out potentially misaligned or lower quality synthetic text samples, we define that at least some ratio $p_f$ of the visual tags exist. For the

---

[2]Or the opposite depending on the user-specified policy

synthetic image, we run it through the VTM again and output the visual tags, then we do the same check against the starting node text ($1b$ or $1d$). Later in Section 4.4, we will show that self-filtering improves the synthetic samples.

## 4 EXPERIMENTS

### 4.1 SETUP

**Tasks and Datasets.** We adopt the CLIP (Radford et al., 2021) model architecture for multimodal representation learning. For pretraining CLIP models, we use two public image-text datasets: CC3M (Sharma et al., 2018) and CC12M (Changpinyo et al., 2021). To evaluate the representation quality of pretrained CLIP models, we measure the zero-shot performance on classification, retrieval, and compositional reasoning tasks. For image classification, we use 25 common vision datasets, including five ImageNet (Deng et al., 2009; Recht et al., 2019) variants and the tasks from the VTAB benchmark (Zhai et al., 2020). We list the detailed dataset information in Appendix A.2. We use COCO (Lin et al., 2014) and Flickr30k (Plummer et al., 2015) for image-to-text and text-to-image retrieval tasks and report the metrics in recall@1. SugarCrepe (Hsieh et al., 2023) is a recent benchmark that measures the compositional understanding of vision-language models, we report the zero-shot accuracy numbers. Additionally, to study the effects of CtrlSynth on long-tail tasks, we evaluate the task accuracy of Places-LT and ImageNet-LT datasets (Liu et al., 2019) by augmenting the tail classes with CtrlSynth synthetic data.

**Training and Baselines.** Note that CtrlSynth itself does not require any training. We conduct pretraining experiments on CLIP models to evaluate the quality of synthetic data. We use ViT-B/16 (Dosovitskiy et al., 2020) architecture for the CLIP vision backbone. For a fair comparison, we train all models for the same number of iterations on the original dataset (baseline) and the dataset mixed with CtrlSynth augmented samples. We use CtrlSynth-cap to denote the original image and synthetic text pair $(1a, 1d)$ from synthesis path $SP(1)$. CtrlSynth-img stands for the synthetic image and original text pair $(1b, 1c)$ from synthesis path $SP(4)$. CtrlSynth-capimg means the synthetic image and text pair $(1d, 1c)$ from synthesis path $SP(3)$. We define CtrlSynth-mix as taking one image-text pair from CtrlSynth-cap and another from CtrlSynth-capimg. We do not take CtrlSynth-img image-text pairs since we found the original texts are noisy and thus a substantial portion of synthetic images are bad quality. We refer CtrlSynth-mix as the default setting unless otherwise stated. We list detailed information in Appendix A.3.

**CtrlSynth Models.** For the VTM, we adopt a hybrid approach by default, we combine the tags from a captioning plus tag extraction pipeline and an advanced multi-label image classifier. We use a recent vision foundation model called Florence-large (Xiao et al., 2024) to generate detailed image descriptions and then extract the objects, attributes, and relations using the Qwen2-7B-Instruct (Yang et al., 2024a) LLM. Then we use an accurate image classifier, the huge variant of CatLIP (Mehta et al., 2024b), to output multiple high-confidence objects and attributes. We show later in Section 4.5 that this hybrid VTM provides the best visual tags compared with using individual approach alone. For the LLM, we use Mistral-NeMo-instruct model (AI, 2024) by default due to its strong instruction-following capability. We choose the stable-diffusion-xl-base-1.0 (Podell et al., 2024) for the text-to-image model by default. We describe the detailed setup in Appendix A.4. In Section 4.5, we study different pretrained models for each of the three modules in CtrlSynth.

### 4.2 MAIN RESULTS

**Image Classification Evaluation.** We conduct the zero-shot evaluation for image classification tasks. Table 1 shows the results across 20 commonly used vision datasets and Table 2 shows the results of 6 ImageNet-related datasets. Notably, CtrlSynth outperforms the baseline consistently by 2.5% to 9.4% for the CLIP models trained on the CC3M and CC12M datasets. We observe that CtrlSynth significantly improves the zero-shot performance (by over 7.7%) by augmenting smaller datasets like CC3M, while the performance gains become smaller on larger datasets like CC12M.

**Image-Text Retrieval Evaluation.** We evaluate the zero-shot image-text retrieval performance for our CtrlSynth and baseline CLIP models and present the recall@1 results in Table 3. CtrlSynth substantially improves the text-to-image and image-to-text retrieval recall by up to 24% and 36% for the Flickr dataset, and overall improves recall by 23.4% on average for CC3M models. CtrlSynth

Table 1: Comparison of the zero-shot classification accuracy between the baseline and CtrlSynth. We report top-1 accuracy for 20 commonly used downstream vision datasets, including 12 tasks in the VTAB benchmark (Zhai et al., 2020) and 8 other ones.

| Data \ Model | CC3M | | CC12M | |
| --- | --- | --- | --- | --- |
| | CLIP | CtrlSynth | CLIP | CtrlSynth |
| CIFAR-10 | 41.5 | **70.3** | 75.4 | **82.6** |
| CIFAR-100 | 14.1 | **34.5** | 47.5 | **53.4** |
| CLEVR Counts | 7.1 | **11.7** | 15.2 | **22.1** |
| CLEVR Distance | 16.1 | **19.8** | **18.6** | 18.0 |
| Caltech-101 | 43.8 | **68.0** | **76.5** | 76.2 |
| Country211 | 0.4 | **0.6** | 1.1 | **1.3** |
| DTD | 11.6 | **17.9** | 23.5 | **29.1** |
| EuroSAT | 12.5 | **15.1** | 25.4 | **27.2** |
| FGVC Aircraft | **1.3** | 0.8 | 0.7 | **1.8** |
| Food-101 | 9.5 | **23.1** | 53.4 | **61.0** |
| GTSRB | 4.6 | **9.7** | 14.5 | **19.1** |
| KITTI | **30.2** | 19.5 | 33.9 | 33.9 |
| Oxford Flowers | 10.8 | **24.8** | 34.5 | **38.9** |
| Oxford-IIIT Pet | 3.1 | **7.9** | 8.0 | **9.4** |
| PatchCamelyon | **50.0** | 48.6 | **52.7** | 50.4 |
| RESISC45 | 17.7 | **27.6** | 36.7 | **39.5** |
| STL-10 | 70.4 | **90.4** | 92.8 | **94.0** |
| SUN397 | 30.7 | **44.3** | 54.1 | **58.1** |
| SVHN | **12.2** | 6.8 | 10.6 | **14.0** |
| Stanford Cars | 0.6 | 0.6 | **2.3** | 2.0 |
| Average | 19.4 | **27.1 (+7.7)** | 33.9 | **36.6 (+2.5)** |

Table 2: Zero-shot top-1 accuracy between the baseline and CtrlSynth on 6 ImageNet datasets.

| Data \ Model | CC3M | | CC12M | |
| --- | --- | --- | --- | --- |
| | CLIP | CtrlSynth | CLIP | CtrlSynth |
| ImageNet-1K | 20.2 | **25.3** | 39.6 | **41.2** |
| ImageNet-V2 | 11.0 | **20.7** | 34.0 | **35.5** |
| ImageNet-S | 3.5 | **12.4** | 28.3 | **33.8** |
| ImageNet-A | 3.0 | **6.5** | 12.0 | **14.9** |
| ImageNet-O | 18.6 | **30.7** | 44.2 | **45.9** |
| ImageNet-R | 11.6 | **28.4** | 47.6 | **55.1** |
| Average | 11.3 | **20.7 (+9.4)** | 34.3 | **37.7 (+3.4)** |

Table 3: Zero-shot retrieval evaluation on the Flickr and COCO datasets. We report the recall@1 numbers. I2T means image-to-text retrieval, and T2I denotes text-to-image retrieval.

| Data \ Model | CC3M | | CC12M | |
| --- | --- | --- | --- | --- |
| | CLIP | CtrlSynth | CLIP | CtrlSynth |
| COCO I2T | 10.9 | **32.3** | 40.5 | **49.8** |
| COCO T2I | 7.6 | **19.8** | 26.7 | **32.2** |
| Flickr I2T | 21.3 | **57.3** | 65.5 | **77.2** |
| Flickr T2I | 14.8 | **39.0** | 48.9 | **58.2** |
| Average | 13.7 | **37.1 (+23.4)** | 45.4 | **54.4 (+9.0)** |

Table 4: We evaluate the compositional reasoning accuracy on the SugarCrepe (Hsieh et al., 2023) benchmark.

| Data | Model | ADD | | REPLACE | | | SWAP | | AVERAGE |
| --- | --- | --- | --- | --- | --- | --- | --- | --- | --- |
| | | Attribute | Object | Attribute | Object | Relation | Attribute | Object | |
| CC3M | CLIP | **69.2** | 71.0 | 69.3 | 80.3 | 55.2 | 52.6 | 50.6 | 64.0 |
| | CtrlSynth | 66.2 | 71.0 | **73.1** | **82.8** | **59.5** | **67.4** | **59.6** | **68.5 (+4.5)** |
| CC12M | CLIP | 70.7 | 77.8 | 78.7 | **88.4** | 66.7 | 61.7 | 62.0 | 72.3 |
| | CtrlSynth | **71.7** | **78.7** | **82.6** | 88.3 | **69.3** | **72.7** | **63.7** | **75.3 (+3.0)** |

also brings over 9% retrieval gains for CC12M models on average. The improvements show that data samples from CtrlSynth have better coverage of visual concepts.

**Compositional Reasoning Results.** A key strength in CtrlSynth is the inclusion of visual tags that contain objects, attributes and relations from an image. To understand how the fine-grained visual attributes and relations affect visual reasoning performance, we evaluate CtrlSynth and baseline on the SugarCrepe (Hsieh et al., 2023) benchmark which measures the compositional reasoning capability of vision language models. We present the results in Table 4. CtrlSynth improves the baseline CLIP compositional reasoning by a large margin (4.5% for CC3M and 3% for CC12M on average). Note that most of the improvements come from the attribute and relation forms in the REPLACE and SWAP categories, for example, CtrlSynth on CC3M improves the REPLACE relation accuracy by 4.3% and SWAP attribute by 14.8%, indicating CtrlSynth models are robust to the attribute and relation changes.

**Comparison with Prior Work.** CtrlSynth pipeline is flexible and supports synthesizing data from different paths. Previous work like VeCLIP (Lai et al., 2024) and LaCLIP (Fan et al., 2023) synthesizing new texts for the images by improving the captions. Though it is impossible to have a completely fair comparison with them[3], the synthetic texts from the synthesis path (2) in CtrlSynth provide similar effects. We present the results on CLIP ViT/B16 models trained on CC3M for the tasks reported in each work. Table 5 shows that CtrlSynth outperforms VeCLIP on most VTAB datasets and improves zero-shot accuracy by 4.8% on average. CtrlSynth also surpasses VeCLIP by 7.9% on the ImageNet 1K dataset. We observe a similar trend when comparing CtrlSynth with LaCLIP in Table 6. Specifically, CtrlSynth achieves an average of 3.4% better accuracy than LaCLIP on 15 common datasets and 2.3% better accuracy on ImageNet 1K.

---

[3]Factors that prohibit apple-to-apple comparison include training software, variations of CC3M samples due to missing images, exact hardware set up, etc.

Table 5: Comparison of the zero-shot classification accuracy between VeCLIP (Vasu et al., 2024) and CtrlSynth for CLIP trained on the CC3M. We report top-1 accuracy (%) for the VTAB benchmark (Zhai et al., 2020) across 9 tasks (6 from natural and 3 from specialized sets). We highlight the best numbers in **bold**.

| Model | Natural Sets | | | | | | Specialized Sets | | | Average | ImageNet 1K |
|---|---|---|---|---|---|---|---|---|---|---|---|
| | Caltech101 | CIFAR100 | SVHN | DTD | OxPet | Flowers102 | EuroSAT | RESISC45 | Camelyon | | |
| CLIP | 39.50 | 9.83 | **20.89** | 7.42 | 7.44 | 10.40 | 11.94 | 7.93 | 50.65 | 18.45 | 5.46 |
| VeCLIP | 54.30 | 17.74 | 18.74 | 11.23 | **10.09** | **22.75** | 7.35 | 16.54 | **52.52** | 23.48 | 15.98 |
| CtrlSynth | **66.10** | **34.09** | 17.66 | **16.76** | 7.77 | 15.55 | **20.83** | **24.59** | 50.79 | **28.24** | **23.82** |

Table 6: We report the zero-shot performance on ImageNet 1K and 15 common downstream datasets for both LaCLIP (Fan et al., 2023) and CtrlSynth for CLIP trained on CC3M. We highlight the best numbers in **bold**.

| Model | Food-101 | CIFAR-10 | CIFAR-100 | SUN397 | Cars | Aircraft | DTD | Pets | Caltech-101 | Flowers | STL-10 | EuroSAT | RESISC45 | GTSRB | Country211 | Average | ImageNet |
|---|---|---|---|---|---|---|---|---|---|---|---|---|---|---|---|---|---|
| CLIP | 10.3 | 54.9 | 21.8 | 25.0 | 0.8 | 1.4 | 10.5 | 12.8 | 43.3 | 10.2 | 77.6 | 14.1 | 19.1 | 6.9 | 0.6 | 20.6 | 15.8 |
| LaCLIP | 14.2 | 57.1 | 27.5 | 35.1 | **1.6** | **1.6** | 16.6 | **15.6** | 52.7 | 14.7 | 86.2 | 15.0 | 24.3 | 6.4 | **1.0** | 24.6 | 21.5 |
| CtrlSynth | **17.8** | **69.5** | **34.1** | **44.9** | 0.7 | 1.2 | **16.8** | 7.8 | **66.1** | **15.5** | **88.3** | **20.8** | **24.6** | **10.9** | 0.7 | **28.0** | **23.8** |

Table 7: Long-tail accuracy on the ImagetNet-LT and Places-LT datasets for the baseline and CtrlSynth models.

| Model | ImageNet-LT | | | | Places-LT | | | |
|---|---|---|---|---|---|---|---|---|
| | Overall | Tail | Medium | Head | Overall | Tail | Medium | Head |
| Baseline | 60.8 | 13.8 | 56.7 | **82.6** | 34.9 | 8.2 | 31.3 | **53.7** |
| CtrlSynth | **66.2 (+5.4)** | **35.1 (+21.3)** | **62.8 (+6.1)** | 81.4 | **38.6 (+3.7)** | **24.4 (+16.2)** | **34.6 (+3.3)** | 51.2 |

## 4.3 Performance on Long-tail Tasks.

Real-world data often have long-tail distributions. Much recent research (Shi et al., 2024; Liu et al., 2019) has focused on developing new learning methods for long-tail recognition tasks. Data augmentation remains an important solution, especially when the tail classes only have a few samples. In this section, we evaluate the effectiveness of synthetic samples from CtrlSynth for long-tail tasks.

**Setup.** We conduct experiments on the ImageNet-LT (Liu et al., 2019) and Places-LT (Liu et al., 2019) datasets. ImageNet-LT is a subset of the original ImageNet-2012 (Deng et al., 2009) and contains 115.8K images from 1000 classes, with 5 to 1280 images per class. Places-LT is even more imbalanced and contains 62.5K images from 365 classes, with 5 to 4980 images per class. The test sets of both datasets are balanced. Following the same setup in (Liu et al., 2019), we report the overall accuracy as well as the accuracy across the head (>100 images), medium (20~100), and tail (<20) classes. We take the same baseline in (Shi et al., 2024) and fine-tune the classifier head of a pretrained CLIP model (ViT-B/16) for 10 epochs (or the same number of iterations for CtrlSynth). For CtrlSynth synthetic samples, we choose the synthetic path $SP(3)$ to generate synthetic images for the tail classes. We mix the CtrlSynth image samples with the original training set of each dataset. We describe more details in Appendix A.2.

**Key Results.** Table 7 shows that CtrlSynth improves the tail class accuracy by 21.3% on ImageNet-LT and by 16.2% on Places-LT. Synthetic samples from CtrlSynth also improve the overall and medium class accuracy by 3~6%, though slightly decrease the head class accuracy.

## 4.4 Analysis

**Data-Efficiency of CtrlSynth in Training CLIP.** To study the data efficiency of CtrlSynth samples, we plot the top1 zero-shot accuracy of the ImageNet validation set in Section 4.3 for the baseline and CtrlSynth CLIP models trained on CC3M. CtrlSynth

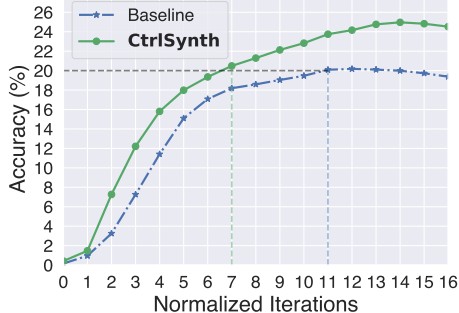

Figure 5: Data efficiency comparison between baseline and CtrlSynth for pretraining CLIP models on CC3M. We normalize the iterations by dividing the total iterations with checkpoint steps.

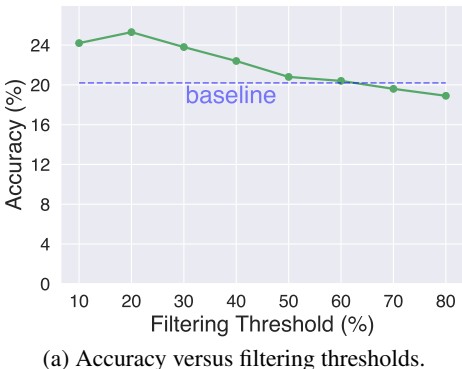 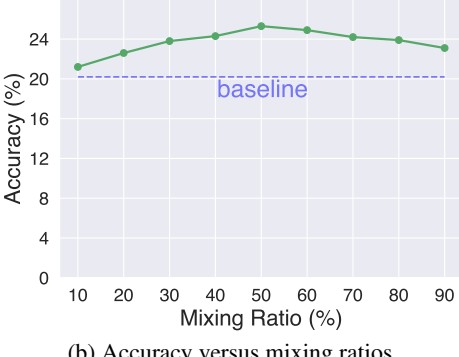

(a) Accuracy versus filtering thresholds.  (b) Accuracy versus mixing ratios.

Figure 6: Study of different filtering thresholds and mixing ratios of CtrlSynth samples. The accuracy numbers are top1 zero-shot accuracy on the ImageNet-1K validation set. The CLIP models are trained on the CC3M dataset and CtrlSynth samples.

reaches the 20% accuracy with 40% fewer iterations than the baseline, indicating that using CtrlSynth samples is more data-efficient. Furthermore, our method can be combined with previous techniques that perform deduplication, filtering, and pruning (Mahmoud et al., 2024; Abbas et al., 2023; Zhang et al., 2024) to further improve data efficiency.

**Statistics and visualization of CtrlSynth Samples.**    In this section, we provide the statistics for the synthetic samples from CtrlSynth. We observe that the text samples from CtrlSynth are usually longer and contain richer information about the image. On average, CtrlSynth texts have over 60 words while original captions contain 8 words. We plot the histogram of the number of words in Figure 7 at Appendix A.6 and visualize examples of CtrlSynth images and texts compared with the original real samples in Figure 8 at Appendix A.6.

**Effects of Self-Filtering.**    CtrlSynth provides off-the-shelf self-filtering to control the quality of synthetic samples. We study the effects of applying different filtering thresholds $p_f$ for the synthetic text and image. We set the same filtering thresholds for both synthetic text and image samples. Intuitively, a higher threshold filters out more synthetic samples thus providing better quality samples that align with original real samples. On the contrary, a lower threshold keeps relatively less aligned samples but encourages more diverse samples. Section 4.4 plots the zero-shot accuracy numbers of CLIP model on ImageNet under different threshold settings, we show that thresholds 10%∼30% provide similar accuracy numbers and setting the filtering threshold to 20% provides the best accuracy. Thresholds higher than 50% do not provide accuracy gains, likely because the aligned synthetic samples lack diversity and fail to augment the original samples.

**Mixing Ratios of Synthetic Samples.**    To better understand how the synthetic image text samples improve CLIP model training, we study different ratios ($p_r$) of mixing CtrlSynth samples with original real ones. During CLIP training, we randomly sample the original sample with probability $0 < p_r < 1$ and our sample with $1 - p_r$. Section 4.4 shows that even adding a small portion ($< 20\%$) of CtrlSynth samples improves the zero-shot accuracy while mixing with 50% provides best accuracy gains. Further higher mixing ratios show diminishing improvements though still better than the baseline that uses all real data.

### 4.5 Ablation Study

In this section, we evaluate the effectiveness of visual tags, the impact of using different pretrained models in the CtrlSynth pipeline, and mixing and filtering effects for CtrlSynth samples. We use the same text and image control policy described in Section 3.2 for all settings. We experiment with CC3M dataset for CLIP pretraining and report the accuracy on the SugarCrepe benchmark, zero-shot accuracy of common downstream vision tasks (same tasks in Table 1), and top1 accuracy on the ImageNet 1k validation set.

Table 8: Evaluation of using different models, visual tags, and synthetic samples in CtrlSynth. '-' denotes the same value from the last row (default setting).

| Study | Model | Tags | Samples | Common Tasks | ImageNet-1K | SugarCrepe |
|---|---|---|---|---|---|---|
| Models | Qwen2-7B, SDXL | - | - | 24.7 | 23.5 | 65.1 |
| | Qwen2-7B, SD3M | - | - | 26.1 | 23.8 | 65.2 |
| | Mistral-Nemo, SD3M | - | - | 26.6 | 25.1 | 68.1 |
| Tags | - | Obj | - | 26.4 | 24.7 | 64.3 |
| | - | Obj+Attr | - | 26.2 | 24.8 | 65.4 |
| Samples | - | - | CtrlSynth-cap, SP(1) | 26.2 | 24.5 | 67.2 |
| | - | - | CtrlSynth-img, SP(4) | 22.1 | 21.8 | 64.4 |
| | - | - | CtrlSynth-capimg, SP(3) | 26.5 | 24.8 | 67.5 |
| CtrlSynth | Mistral-Nemo, SDXL | Obj+Attr+Rel | CtrlSynth-mix | 27.1 | 25.3 | 68.5 |

**Different Pretrained Models.** We choose an alternate LLM and a different text-to-image model to understand how different pretrained models affect the quality of synthetic samples. CtrlSynth pipeline is flexible so we can easily swap the pretrained LLM and text-to-image models. Specifically, we use Qwen2-7B (Yang et al., 2024a) for the LLM and Stable Diffusion 3 Medium (Esser et al., 2024) (SD3M) for the text-to-image model. Comparing the first and last rows in Table 8, we find using a smaller LLM like Qwen2-7B degrades the task performance on all three tasks, indicating that using a strong LLM is key to synthesizing high quality texts. The accuracy boost (+3%) on SugarCrepe benchmark shows the LLM is effective in recombining the visual tags in a compositional way to form diverse synthetic texts. We also point out that using a more recent diffusion model like SD3M provides similar task performance numbers, this is likely because SD3M has fewer (2B versus 3.5B) parameters compared to SDXL, limiting the image generation capability.

**Effectiveness of Visual Tags.** We study the effects of using different categories of visual tags, *i.e.*, using only objects (Obj), objects plus attributes (Obj+Attr), and all categories including relations (Obj+Attr+Rel). In Table 8, comparing the second and last row, we show attributes marginally improve the CLIP performance on compositional reasoning but not much on zero-shot vision tasks. Importantly, visual relations improves the performance on all three tasks, and significantly improves compositional reasoning performance by over 4%.

**CtrlSynth Samples from Different Synthesis Paths.** CtrlSynth pipeline supports synthesizing images or texts from different paths, we evaluate their quality by measuring the downstream task accuracy of the CLIP models trained on them. The penultimate and last rows in Section 4.5 show all CtrlSynth samples provides performance gains on downstream tasks, except the CtrlSynth-img samples where they do not improve compositional reasoning performance. CtrlSynth-img samples have the least augmentation benefits and are likely due to the original real texts are noisy and thus the generated images are not of high quality. Notably, mixing with synthetic captions (CtrlSynth-cap, CtrlSynth-capimg, and CtrlSynth-mix) provides meaningful augmentation benefits, highlight the importance of using LLMs to recombine the visual tags.

## 5 CONCLUSION

Synthetic data emerges as a viable solution to address challenges in curating high-quality samples from noisy, misaligned, and long-tail web data. However, existing data synthesis pipelines are rigid and the generation process is hard to control and thus being tailored for ad hoc use cases. We develop CtrlSynth, a new image-text synthesis pipeline that allows users to control the data generation in a fine-grained way. CtrlSynth decomposes the semantics of images and texts into basic elements and uses pretrained foundation models to recompose them based on specified control policies. This way, CtrlSynth provides flexible and diverse image-text samples. Synthetic samples from CtrlSynth improve the long-tail task performance by a large margin. They also significantly boost the zero-shot and compositional capability of CLIP models and enable data-efficient multimodal learning.

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

# A APPENDIX

## A.1 CONTROL POLICIES

**Text Prompt Templates.** We provide example control policies for text synthesis as predefined prompt templates, the first five templates do not include original text:

1. "Create a detailed and high-quality caption using phrases that represent the entities or objects, their unique attributes, and the visual relationships in the scene depicted. Phrases: {phrases}."

2. "Compose a rich and immersive caption by incorporating a set of phrases that illustrate the entities or objects, their defining attributes, and the interconnections presented within the image. Phrases: {phrases}."

3. "Formulate an articulate and informative caption by using a series of phrases that outline the entities, their attributes, and their visual relationships depicted in an image. Phrases: {phrases}."

4. "Using a set of phrases that highlight the entities, attributes, and their visual associations in an image, craft a detailed and expressive caption. Phrases: {phrases}."

5. "Construct a comprehensive and expressive caption by integrating phrases that detail the entities, their features, and the spatial or thematic relationships in an image. Phrases: {phrases}."

The following five templates include the original text, which is useful for maintaining the original meaning:

1. "Create a comprehensive caption that faithfully represents the objects, attributes, and their relationships contained within the provided sentence and phrases. Given sentence: {caption}. Given phrases: {phrases}. If the original caption specifies particular give phrases, maintain their integrity while using the phrases to enhance the description."

2. "Write a faithful caption by integrating the given phrases with the original sentence. Given sentence: {caption}. Given phrases: {phrases}. Ensure any objects or specific nouns from the original caption are preserved while elaborating on the visual relationships and attributes provided in the phrases to create a more detailed depiction."

3. "Provide a faithful and informative image caption using a given sentence and few phrases. Sentence: {caption}, phrases: {phrases}. Consider the initial sentence as a base for the overall context and ensure that specific objects or nouns such as numbers, car models, animals, etc., are preserved in the new caption. Integrate the given phrases, which describe entities, attributes, or visual relationships, to enrich and elaborate on the original meaning. Maintain fidelity to the original content while enhancing descriptive quality."

4. "Make a detailed caption based on the given phrases and a given sentence. Given phrases: {phrases}. Given sentence: {caption}. The sentence serves as a foundation, while the

phrases elaborate on elements depicted in the image, like objects, their characteristics, and interactions. Preserve any pivotal information concerning objects, attributes, and their relations present in the sentence."

5. "Write a new faithful and high-quality caption based on the given phrases and a given sentence. The given sentence is the original caption and the phrases are entities or objects, attributes, and their visual relationships in an image. Given sentence: {caption}. Given phrases: {phrases}. If the sentence contains objects or nouns (e.g. digits, car models, planes, pets, animals, etc.), the new caption should be faithful and keep this information. Otherwise, use the phrases to create the new caption."

**Image Prompt Templates.** We provide five image prompt templates:

1. "real": "a real photo. {prompt}. 35mm photograph, film, bokeh, professional, 4k, highly detailed",

2. "nocap": "a real photo showing {prompt}. highly detailed"

3. "isometric": "isometric style {prompt} . vibrant, beautiful, crisp, detailed, ultra detailed, intricate"

4. "enhance": "breathtaking {prompt}. award-winning, professional, highly detailed"

5. "quality": "masterpiece, best quality, ultra detailed, {prompt}. intricate details"

## A.2 DATASETS DETAILS

**Evaluation Datasets.** We list the vision datasets for evaluation in Table 9.

Table 9: Details of evaluation datasets.

| Dataset | Metric | Classes | Test Set Size |
|---|---|---|---|
| CIFAR-10 (Krizhevsky, 2009) | Accuracy | 10 | 10000 |
| CIFAR-100 (Krizhevsky, 2009) | Accuracy | 100 | 10000 |
| CLEVR Counts | Accuracy | 8 | 15000 |
| CLEVR Distance | Accuracy | 6 | 15000 |
| Caltech-101 (Fei-Fei et al., 2006) | Mean Per Class Recall | 102 | 6085 |
| Country211 (Radford et al., 2021) | Accuracy | 211 | 21100 |
| DTD (Cimpoi et al., 2014) | Accuracy | 47 | 1880 |
| EuroSAT (Helber et al., 2018) | Accuracy | 10 | 5400 |
| FGVC Aircraft (Maji et al., 2013) | Mean Per Class Recall | 100 | 3333 |
| Food-101 (Bossard et al., 2014) | Accuracy | 101 | 25250 |
| GTSRB (Stallkamp et al., 2011) | Accuracy | 43 | 12630 |
| KITTI (Geiger et al., 2013) | Accuracy | 4 | 711 |
| Oxford Flowers-102 (Nilsback & Zisserman, 2008) | Mean Per Class Recall | 102 | 6149 |
| Oxford-IIIT Pet (Parkhi et al., 2012) | Mean Per Class Recall | 37 | 3669 |
| PatchCamelyon (Veeling et al., 2018) | Accuracy | 2 | 32768 |
| RESISC45 (Cheng et al., 2017) | Accuracy | 45 | 6300 |
| STL-10 (Coates et al., 2011) | Accuracy | 10 | 8000 |
| SUN397 (Xiao et al., 2010) | Accuracy | 397 | 108754 |
| SVHN (Netzer et al., 2011) | Accuracy | 10 | 26032 |
| Stanford Cars (Krause et al., 2013) | Accuracy | 196 | 8041 |
| ImageNet-1K (Deng et al., 2009) | Accuracy | 1000 | 50000 |
| ImageNet-V2 (Recht et al., 2019) | Accuracy | 1000 | 10000 |
| ImageNet-S (Wang et al., 2019) | Accuracy | 1000 | 50889 |
| ImageNet-A (Hendrycks et al., 2021b) | Accuracy | 200 | 7500 |
| ImageNet-O (Hendrycks et al., 2021b) | Accuracy | 200 | 2000 |
| ImageNet-R (Hendrycks et al., 2021a) | Accuracy | 200 | 30000 |
| Flickr (Plummer et al., 2015) | Mean Recall@1 | - | 1000 |
| MSCOCO (Lin et al., 2014) | Mean Recall@1 | - | 5000 |

**Long-tail Datasets.** For the tail classes in ImageNet-LT and Places-LT, we generate synthetic images using the "real" style of image prompt template, and we generate 7 samples per tail class so

Table 10: Training hyper-parameters.

(a) Pretraining CLIP on CC3M and CC12M.

| Hyperparameter | CC3M | CC12M |
|---|---|---|
| Total iterations | 56,429 | 55,429 |
| Warmup iterations | 2822 | 2771 |
| Image size | 224 | 224 |
| LR scheduler | Cosine | Cosine |
| Max. LR | 0.002 | 0.002 |
| Min. LR | 0.00002 | 0.00002 |
| Optimizer | AdamW | AdamW |
| AdamW $\beta$'s | (0.9, 0.98) | (0.9, 0.98) |
| Weight decay | 0.2 | 0.2 |
| Batch size per GPU | 256 | 256 |
| # A100 GPUs | 8 | 32 |
| A100 GPU Memory | 40 GB | 40 GB |

(b) Finetuning CLIP on Places-LT and ImageNet-LT.

| Hyperparameter | Places-LT | ImageNet-LT |
|---|---|---|
| Total Iterations | 56,429 | 55,429 |
| Warmup Iterations | 2822 | 2771 |
| Image size | 224 | 224 |
| Loss type | CrossEntropy | CrossEntropy |
| LR scheduler | Cosine | Cosine |
| Learning rate | 0.01 | 0.01 |
| Optimizer | SGD | SGD |
| Momentum | 0.9 | 0.9 |
| Weight decay | 5e-4 | 5e-4 |
| Batch size per GPU | 128 | 128 |
| # A100 GPUs | 1 | 1 |
| A100 GPU Memory | 40 GB | 40 GB |

that we roughly double the size of the original real datasets. We obtain 80.4k synthetic samples for ImageNet-LT and 55.2K for Places-LT.

## A.3 TRAINING DETAILS

**Pretraining Hyper-parameters.** We pretrain the CLIP for the same number of iterations for both the baseline and CtrlSynth. For example, suppose we train for $E$ epochs, if the original dataset has $N$ samples, CtrlSynth has generated $N'$ samples ($N' <= N$ due to filtering), then the total samples are $E * N$, we train CtrlSynth models for $\frac{E*N}{N+N'}$ epochs. This guarantees that the baseline and CtrlSynth CLIP models have seen the same number of data samples.

Table 10 lists the hyper-parameters used for pretraining on CC3M and CC12m. We use AdamW (Loshchilov & Hutter, 2018) with default $\beta$ values as an optimizer and binary cross-entropy loss as an objective function. We use cosine learning rate schedule (Loshchilov & Hutter, 2022). We use the CoreNet library (Mehta et al., 2024a; 2022) for all pretraining experiments. We adapt the LIFT codebase (Shi et al., 2024) for fine-tuning long-tail tasks, main modifications include adding support for iteration-based training and data loader for multiple datasets.

## A.4 CTRLSYNTH INFERENCE DETAILS

**VTM.** We use a hybrid tagging model consisting of two stages. We first run the ViT-Huge variant of CatLIP (Mehta et al., 2024b) for each image and output top20 classes based on the sigmoid score of prediction logits, then we convert the class indices to actual word labels. The vocabulary size of CatLIP is 24320. Most of the vocabulary words are nouns and single-word attributes. We then run the Florence-large (Xiao et al., 2024) for each image to extract detailed captions using the task prompt `<MORE_DETAILED_CAPTION>`. After that, we run Qwen2-7B-Instruct (Yang et al., 2024a) to extract objects, attributes, and relations from the Florence captions. We then merge the objects field with CatLIP-predicted labels. The extraction instruction contains a 2-shot example and we list the prompt template below:

```
For a given image caption, identify all the attributes, objects or entities, and visual
    relationships or actions that are phrases. The phrases should only come from the
    caption. Separate the phrases by comma without formatting. Output three lines:
attributes: phrases
objects: phrases
relations: phrases

Examples:

caption: The image is a close-up portrait of a middle-aged man wearing a white cowboy
    hat. He appears to be in his late 60s or early 70s, with gray hair and a serious
    expression on his face. He is wearing a dark suit jacket and a light blue collared
    shirt. The background is a clear blue sky with trees visible in the distance. The
    man is looking off to the side with a slight smile on his lips.
attributes: close-up, middle-aged, white cowboy hat, gray hair, serious expression,
    light blue
```

```
objects: portrait, man, hat, face, dark suit jacket, shirt, blue sky, trees, lips
relations: wearing a, visible in the distance, looking off to the side, slight smile on
    his lips

caption: The image shows a female singer performing on a stage. She is standing on a set
     of stairs with her legs spread apart and holding a microphone in her hand. The
    stage is lit up with red and blue lights and there is a large circular screen in
    the background. The singer is wearing a black and white patterned outfit with high
    heels. She appears to be in the middle of a song or performance.
attributes: female singer, stage, set of stairs, red and blue lights, large circular
    screen, black and white patterned outfit, high heels
objects: female singer, stage, set of stairs, legs, microphone, screen, outfit, high
    heels, song, performance
relations: performing on a stage, standing on, her legs spread apart, holding, lit up,
    background, wearing, in the middle of a song

caption: {caption}
```

CatLIP is available in CoreNet so we use it directly for inference and we wrap the Florence Transformers (Wolf et al., 2020) code into the CoreNet inference pipeline for easier integration.

**LLM.** We use the vLLM engine (Kwon et al., 2023) for offline inference in Qwen2 and Mistral-Nemo. We use greedy decoding for the generation.

**Text-to-image Model.** We use the diffusers (von Platen et al., 2022) library for diffusion model inference. For both SDXL and SD3M models, we use float16 dtype with a guidance scale of 7.0 and set the diffusion steps to 28.

### A.5 CTRLSYNTH SELF-FILTERING DETAILS

CtrlSynth is a closed-loop system and supports self-filtering for bad-quality synthetic text or image samples. To implement synthetic text filtering, we first compute the percentage of visual tags that appear in the synthetic text compared to the original text, then we filter out the sample if the percentage of visual tags is lower than a predefined threshold $p_f$. We empirically choose $p_f$ based on the zero-shot accuracy of trained CLIP models evaluated on the ImageNet validation set. Similarly, to filter synthetic images, we first extract the visual tags of the synthetic images by running them through VTM, then compute the percentage of visual tags in the original image and filter out image samples if the percentage is lower than $p_f$.

### A.6 MORE ANALYSIS DETAILS

**CtrlSynth Samples.** For CC3M, the original dataset has 2.8 million image-caption pairs, CtrlSynth-cap contains 2.6 million captions, CtrlSynth-img contains 2.4 million images, and CtrlSynth-mix contains 5.1 million image-caption pairs. Original CC12M has 11.3 million image-caption samples, CtrlSynth-cap consists of 10.2 million captions, CtrlSynth-img contains 9.5 million images, and CtrlSynth-mix has 19.7 million image-caption pairs.

**CtrlSynth Synthetic Texts.** We plot the number of words for synthetic texts generated by CtrlSynth and compare them with original real texts in Figure 7.

**Visualization.** We show examples of CtrlSynth images and texts compared with the original real samples in Figure 8.

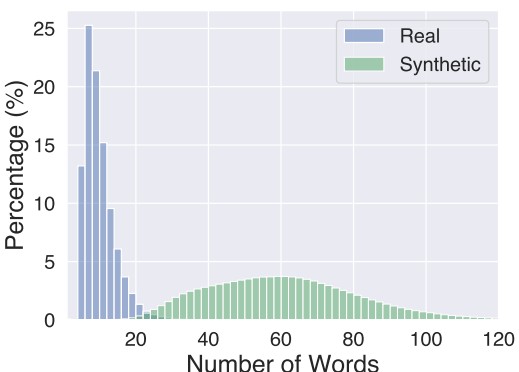

Figure 7: Number of words for the original captions and CtrlSynth synthetic texts on CC3M.

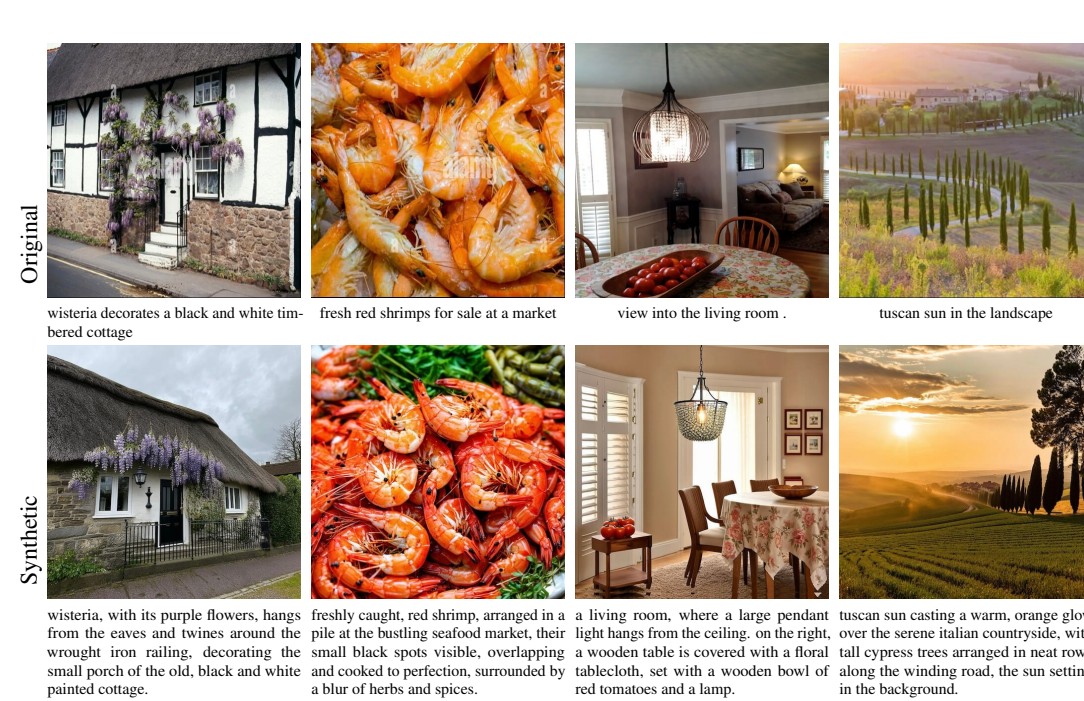

Figure 8: Randomly selected CC3M examples of real images and captions (the first row) with their corresponding CtrlSynth synthetic samples (the second row).

