# OpenReview forum: "CtrlSynth: Controllable Image Text Synthesis for Data-Efficient Multimodal Learning"
_ICLR.cc/2025/Conference — Submitted to ICLR 2025_

### Official Review · Reviewer_JT2Z · 2024-10-20

**Soundness:** 3
**Presentation:** 3
**Contribution:** 3
**Rating:** 5
**Confidence:** 4

**Summary:**

This paper introduces CtrlSynth, a controllable pipeline for generating synthetic image-text data to improve multimodal models. By allowing fine-grained control over data synthesis, CtrlSynth decomposes and recomposes visual semantics using pretrained models, enhancing diversity and alignment of generated samples.

**Strengths:**

- Introduces a new controllable synthesis pipeline (CtrlSynth) that allows fine-grained data manipulation, enabling user-defined policies for image-text synthesis.
- Achieving significant performance improvements across diverse tasks such as zero-shot classification, retrieval, and long-tail recognition is inspiring.
- Clearly explains the methodology with diagrams and examples, easy to understand the synthesis process and its components.

**Weaknesses:**

1. In the ablation experiments, it was observed that the performance improvement brought by CtrlSynth-img alone is minimal. Would it be possible to completely remove the generation of synthetic images and focus resources on improving the diversity and quality of synthetic text? Would this lead to consistent performance improvements across all tasks?

2. The paper mentions that CtrlSynth uses a self-filtering mechanism to improve the quality of synthetic data, but it lacks detailed explanations about the implementation, such as how the alignment threshold for visual tags is selected.

3. The paper does not explain in depth how CtrlSynth fundamentally differs from other caption augmentation methods like VeCLIP and LaCLIP. It is necessary to provide a clearer comparison, clarifying whether the increased diversity brought by the decomposition of visual tags and user control strategies is more important, or whether it is the generation of more fine-grained semantic captions that matters.

4. The experiments may be limited to a few selected models (e.g., Mistral-Nemo and Qwen2-7B). Would using larger LLMs lead to better results?

5. A drawback of this method is that the data generation pipeline involves multiple different models and is not end-to-end in training, requiring substantial resources and time for building the synthetic data in the early stages.

**Questions:**

See weaknesses.

---

> ### Author Response · Authors · 2024-11-19
>
> >In the ablation experiments, it was observed that the performance improvement brought by CtrlSynth-img alone is minimal. Would it be possible to completely remove the generation of synthetic images and focus resources on improving the diversity and quality of synthetic text? Would this lead to consistent performance improvements across all tasks?
>
> **Response 1**:
> Thank you for raising this point. The inclusion of image synthesis in our approach is to enable a broader range of diverse text-image synthesis paths and to complete the closed-loop pipeline. While CtrlSynth-img alone provides only minimal improvements in CLIP performance, it has significant potential to improve results in cases where the text is noisy or when the quality of synthetic text degrades.
> Given the modular nature of CtrlSynth, we can easily remove or replace either component depending on the use case. For CLIP specifically, we could remove CtrlSynth-img and focus on improving the synthetic text. However, for other tasks, such as long-tail vision datasets, we still need synthetic images.
>
> >The paper mentions that CtrlSynth uses a self-filtering mechanism to improve the quality of synthetic data, but it lacks detailed explanations about the implementation, such as how the alignment threshold for visual tags is selected.
>
> **Response 2**:
> In Section 4.4 and Figure 6(a), we describe the methodology and results related to different thresholds for self-filtering. The threshold is selected based on the zero-shot accuracy of the trained CLIP models, evaluated on the ImageNet validation set. To provide further clarity, we have added additional explanations for the self-filtering process in Appendix A.5.
>
> >The paper does not explain in depth how CtrlSynth fundamentally differs from other caption augmentation methods like VeCLIP and LaCLIP. It is necessary to provide a clearer comparison, clarifying whether the increased diversity brought by the decomposition of visual tags and user control strategies is more important, or whether it is the generation of more fine-grained semantic captions that matters.
>
>
> **Response 3**:
> Compared to VeCLIP and LaCLIP, the key difference is that our CtrlSynth system has more fine-grained visual tags and more diverse synthesis paths. In the ablation study (Section 4.5, Table 8), we provide evidence that the improvements of CtrlSynth come from both the use of more fine-grained semantic captions and the diverse text-image synthesis path. We also present the table below:
>
> | Study                             | Model                | Tags         | Samples              | Common Tasks | ImageNet-1K | SugarCrepe |
> |-----------------------------------|----------------------|--------------|----------------------|--------------|-------------|------------|
> | **Models**                        |                      |              |                      |              |             |            |
> |                                   | Qwen2-7B, SDXL       | -            | -                    | 24.7         | 23.5        | 65.1       |
> |                                   | Qwen2-7B, SD3M       | -            | -                    | 26.1         | 23.8        | 65.2       |
> |                                   | Mistral-Nemo, SD3M   | -            | -                    | 26.6         | 25.1        | 68.1       |
> | **Tags**                          |                      |              |                      |              |             |            |
> |                                   | -                    | Obj          | -                    | 26.4         | 24.7        | 64.3       |
> |                                   | -                    | Obj+Attr     | -                    | 26.2         | 24.8        | 65.4       |
> | **Samples**                       |                      |              |                      |              |             |            |
> |                                   | -                    | -            | CtrlSynth-cap, SP(1)       | 26.2         | 24.5        | 67.2       |
> |                                   | -                    | -            | CtrlSynth-img, SP(4)       | 22.1         | 21.8        | 64.4       |
> |                                   | -                    | -            | CtrlSynth-capimg, SP(3)    | 26.5         | 24.8        | 67.5       |
> | **CtrlSynth**                          | Mistral-Nemo, SDXL   | Obj+Attr+Rel | CtrlSynth-mix              | 27.1         | 25.3        | 68.5       |

---

> ### Author Response · Authors · 2024-11-19
>
> >The experiments may be limited to a few selected models (e.g., Mistral-Nemo and Qwen2-7B). Would using larger LLMs lead to better results?
>
> **Response 4**:
> Mistral-Nemo, with 12 billion parameters, is a larger model compared to Qwen2-7B, which has 7 billion parameters. As shown in Table 8, we observe improvements when using the larger Mistral-Nemo model. We leave the exploration of scaling to even larger models for future work.
>
>
> >A drawback of this method is that the data generation pipeline involves multiple different models and is not end-to-end in training, requiring substantial resources and time for building the synthetic data in the early stages.
>
> **Response 5**:
> We believe that the modularity of our pipeline, which supports different models, is a strength rather than a drawback. It does not rely on model-specific configurations, making it flexible and adaptable. Additionally, the pipeline is currently training-free, meaning it does not require end-to-end training. Our focus is on leveraging the capabilities of pretrained models in a plug-and-play manner to facilitate data generation, rather than fine-tuning or adapting the models for specific use cases.

---

### Official Review · Reviewer_nSoW · 2024-10-30

**Soundness:** 3
**Presentation:** 3
**Contribution:** 3
**Rating:** 6
**Confidence:** 4

**Summary:**

This paper introduces a multimodal data synthesis pipeline called CtrlSynth. Specifically, CtrlSynth includes a vision tagging model to extract key objects, attributes, and relations from an image, which can then optionally be combined with the original text for a language model to generate new image descriptions. Finally, the newly generated image caption is input into a text-to-image model to generate an image. The authors have demonstrated the effectiveness of their pipeline by comparing it with CLIP pretraining data. Overall, the enhanced dataset appears to be superior to the original one.

**Strengths:**

1. The idea is clear and effective: by combining multiple expert models, we can obtain fine-grained image tags, captions, and synthetic images, which together help to create a high-quality synthetic dataset.

2. The modularized pipeline is flexible, as each model can be replaced without affecting the performance of the other components.

3. Experiments are comprehensive. Compared to the baseline CLIP, the improvements from CtrlSynth are evident.

**Weaknesses:**

1. Practical concerns: By using several models, such as the vision tagging model, LLM, and diffusion model, the proposed method might not be efficient for scaling up to larger datasets, particularly considering the time cost associated with image synthesis.

2. The assumption behind CtrlSynth is based on a fixed number of data samples, where the method helps a model achieve better performance than training on the original dataset. However, given the recent trends in LLM and multimodal LLM research, where pretraining data continues to scale up, the proposed method may not be scalable for very large datasets. While this is a challenge, under the current setting in the paper, CtrlSynth is indeed effective.

**Questions:**

Can the authors provide details on the overall efficiency of the proposed pipeline? For example, how long does it take to generate 1 million images along with their captions? It would also be good to know the time cost at each component, e.g. vision tagging, caption generation, image generation. A more complete picture of the efficiency in the pipeline would better help to assess the value of this work.

---

> ### Author Response · Authors · 2024-11-19
>
> >Practical concerns: By using several models, such as the vision tagging model, LLM, and diffusion model, the proposed method might not be efficient for scaling up to larger datasets, particularly considering the time cost associated with image synthesis.
> >The assumption behind CtrlSynth is based on a fixed number of data samples, where the method helps a model achieve better performance than training on the original dataset. However, given the recent trends in LLM and multimodal LLM research, where pretraining data continues to scale up, the proposed method may not be scalable for very large datasets. While this is a challenge, under the current setting in the paper, CtrlSynth is indeed effective.
>
> **Response 1**:
> Thank you for acknowledging the effectiveness of our method in the current setting. Our primary goal is to demonstrate the effectiveness of CtrlSynth under a fixed number of data samples, and we have shown that our approach can enhance model performance compared to training solely on the original dataset. We agree that the efficiency of CtrlSynth is constrained by the resources required for image synthesis, particularly as datasets grow larger. Nonetheless, our method provides a valuable step forward within the existing setting, and future work could explore optimizations or alternative strategies to address these scalability concerns.
>
> >Can the authors provide details on the overall efficiency of the proposed pipeline? For example, how long does it take to generate 1 million images along with their captions? It would also be good to know the time cost of each component, e.g. vision tagging, caption generation, image generation. A more complete picture of the efficiency in the pipeline would better help to assess the value of this work.
>
> **Response 2**: we will include detailed efficiency information in the revised paper. Below is a summary of the GPU hours (using H100) required for processing 1 million images and captions:
> - Visual tagging: 89 GPU hours (52 for Florence, 16 for CatLIP, 21 for Qwn2 extraction)
> - Caption generation: 32 GPU hours for running Mistral-Nemo inference
> - Image generation: 4608 GPU hours for running SDXL

---

> > ### Comment · Reviewer_nSoW · 2024-11-25
> > **Official Comments from Reviewer**
> >
> > Thanks for the authors'rebuttal. I have carefully read the comments from other reviewers. For my own comments, I'm still concerned about the practical usage of the proposed pipeline. For example, it requires more than 3 weeks for image generation based on 8 GPUs and 1M images.
> >
> > Besides, I'm a bit curious about Q1 from Reviewer JT2Z, where the reviewer mentioned the performance gain by simply focusing on the the diversity and quality of synthetic text.  Given the fact that this method is not efficient if generating images, I think the text-only performance would be more interesting as it can be much more efficient for the practical usage. However, it seems that the authors do not validate this potential.

---

> > > ### Author Response · Authors · 2024-11-25
> > >
> > > Thank you for the engaging discussion. We understand that image generation demands significantly more GPU resources compared to other synthesis methods. While it might appear optional for certain multimodal use cases, it is necessary for long-tail vision tasks where text synthesis alone falls short. The primary computational bottleneck in image generation lies in the multi-step diffusion process (28 steps in our case). However, techniques like SDXL-Lightning demonstrate that a 4-step diffusion process can produce similar-quality images. Leveraging such methods could potentially reduce total GPU hours from 4608 to just 660. Our immediate goal is to validate image synthesis within the CtrlSynth closed-loop pipeline, and we will include discussions on optimizing synthesis efficiency to support practical deployment scenarios.
> > >
> > > Text-only synthesis is a key component in our CtrlSynth pipeline, and we have validated the effectiveness of our text synthesis methods, specifically, Tables 5 and 6 highlight how our decompose-and-recompose feature for visual tags surpasses previous text-only synthesis methods such as VeCLIP and LaCLIP. Additionally, our ablation study (Table 8) confirms that incorporating visual tags significantly enhances both the diversity and quality of synthetic text.
> > >
> > > Please let us know if you still have concerns, we are happy to make further clarifications.

---

### Official Review · Reviewer_yHqv · 2024-11-03

**Soundness:** 3
**Presentation:** 3
**Contribution:** 2
**Rating:** 3
**Confidence:** 5

**Summary:**

The paper proposes CtrlSynth, a controllable image-text synthesis pipeline for data-efficient multimodal training. Addressing limitations in existing large-scale datasets that are often noisy and misaligned, CtrlSynth enables fine-grained control by decomposing images into basic elements and applying user-specified modifications to synthesize new data. This training-free and flexible pipeline can work with different models and supports closed-loop synthesis (image to text and vice versa). The proposed method also boosts the performance of multimodal model training.

**Strengths:**

- The paper focuses on noise and misalignment in the large-scale image-text datasets, which is a critical challenge in multimodal learning.
- The paper introduces an innovative approach that emphasizes fine-grained control, utilizing generative models to decompose and refine images and texts at a detailed level. Notably, it is training-free and suited for integration with different pre-trained generative models.
- The experiments presented in the paper show that the proposed method improves downstream performances of multimodal models.

**Weaknesses:**

The paper, while contributing valuable ideas, has several notable weaknesses that are significant and need to be addressed.

### Methodological Weaknesses
- The proposed pipeline shares significant similarities with GenArtist[1] on image editing. The paper does not clearly demonstrate the differences between this work and GenArtist. It is important for the authors to specify these distinctions and highlight the novelty of their approach.  Additionally, a thorough comparison should be incorporated into the experimental section to strengthen the evaluation.
- While fine-grained control is presented as the main contribution, represented by the text and image controllers in the pipeline, the design is inadequate and lacks clarity. The design of the pipeline does not effectively demonstrate how the editing condition is provided to the generative model in a fine-grained manner. The text controller relies solely on prompt concatenation, making the mapping between visual tags and policies unclear and limiting precise control. Additionally, the paper does not address how to maintain image consistency after editing, which is essential for practical use. These shortcomings contribute to potential inconsistencies and an insufficient explanation of how fine-grained control is maintained. The image controller exists the same problem.

### Experimental Limitations
- The datasets used (CC3M and CC12M) are relatively small, with no experiments conducted on larger datasets such as LAION-400M or LAION-5B.
- The paper only tests a limited range of multimodal model structures, lacking experiments on models like BLIP and CLIP of different ViT models.
- The study does not address data-efficiency validation. Existing data-efficiency-focused works, such as SemDeDup[2], Filter-&-Align[3], and Sieve[4], refine or filter datasets for better performance. The paper should include comparisons with these approaches in terms of model performance and the amount of training data.

---
Reference

[1] Zhenyu Wang, Aoxue Li, Zhenguo Li, and Xihui Liu. GenArtist: Multimodal LLM as an Agent for Unified Image Generation and Editing. arXiv.2407.05600.

[2] Amro Abbas, Kushal Tirumala, Daniel Simig, Surya Ganguli and Ari S. Morcos. SemDeDup: Data-efficient learning at web-scale through semantic deduplication. arXiv.2303.09540.

[3] Lei Zhang, Fangxun Shu, Tianyang Liu, Sucheng Ren, Hao Jiang, and Cihang Xie. Filter & Align: Leveraging Human Knowledge to Curate Image-Text Data. arXiv.2312.06726.

[4] Anas Mahmoud, Mostafa Elhoushi, Amro Abbas, Yu Yang, Newsha Ardalani, Hugh Leather, and Ari Morcos. Sieve: Multimodal Dataset Pruning Using Image Captioning Models. arXiv.2310.02110.

**Questions:**

Refer to the Weakness.

---

> ### Author Response · Authors · 2024-11-19
>
> >The proposed pipeline shares significant similarities with GenArtist[1] on image editing. The paper does not clearly demonstrate the differences between this work and GenArtist. It is important for the authors to specify these distinctions and highlight the novelty of their approach. Additionally, a thorough comparison should be incorporated into the experimental section to strengthen the evaluation.
>
> **Response 1**:  We appreciate the reviewer’s mention of the GenArtist paper. However, we would like to point out that GenArtist is a concurrent work, classified as contemporaneous according to the ICLR policy (refer to the FAQ section in the ICLR Reviewer Guide, https://iclr.cc/Conferences/2025/ReviewerGuide), and authors are not required to compare their work to the paper.
>
> That said, we have clarified the key distinctions between our work and GenArtist. Specifically: (1) Our CtrlSynth pipeline offers support not only for controllable image synthesis but also for diverse text and image synthesis paths. In Section 3.2, we detail four unique synthesis paths that our method facilitates. (2) While GenArtist focuses on enabling more fine-grained and precise control over image generation, primarily benchmarking against methods like MagicBrush and InstructPix2Pix, our pipeline could benefit from such advancements. Nonetheless, it remains an open research question how to automate the generation of image editing instructions for each dataset sample, given that approaches like MagicBrush and InstructPix2Pix require manually crafted per-sample instructions and require additional training of text-to-image models to support such precise control.
>
> >While fine-grained control is presented as the main contribution, represented by the text and image controllers in the pipeline, the design is inadequate and lacks clarity. The design of the pipeline does not effectively demonstrate how the editing condition is provided to the generative model in a fine-grained manner. The text controller relies solely on prompt concatenation, making the mapping between visual tags and policies unclear and limiting precise control. Additionally, the paper does not address how to maintain image consistency after editing, which is essential for practical use. These shortcomings contribute to potential inconsistencies and an insufficient explanation of how fine-grained control is maintained. The image controller exists the same problem.
>
> **Response 2**:
> We have detailed our design approach in Section 3.1, where we outline the visual tags that enable users to exercise fine-grained control through editing. In line 201 of the revised version, we introduce three control policies for the text controller and two control policies for the image controller. It’s important to clarify that we do not modify the underlying models (LLMs or text-to-image models). Instead, our editing is facilitated solely through input textual instructions.
> While methods like InstructPix2Pix could be explored to maintain image consistency post-editing, applying such techniques at scale remains a significant challenge. Our primary objective is to augment existing datasets by synthesizing diverse samples. Interestingly, we find value in synthetic images that are not perfectly aligned, as they offer beneficial semantic augmentation to the original datasets. In fact, enforcing complete alignment in training can be detrimental, as it restricts the augmentation potential and hampers overall performance.

---

> ### Author Response · Authors · 2024-11-19
>
> >The datasets used (CC3M and CC12M) are relatively small, with no experiments conducted on larger datasets such as LAION-400M or LAION-5B.
>
> **Response 3**:  The main goal of CtrlSynth is to demonstrate the effectiveness and controllability of diverse text-image synthesis across different settings, including image-text datasets like CC3M and CC12M, as well as vision longtail datasets. We acknowledge that our dataset scale is relatively small, but scaling the synthesis pipeline to larger datasets would require substantial computational resources, particularly for image generation.
>
> Moreover, we are unable to utilize the LAION datasets due to the presence of sensitive and NSFW content, as highlighted in recent research ("Into the LAION’s Den: Investigating Hate in Multimodal Datasets," https://arxiv.org/abs/2311.03449). This poses legal challenges that prevent us from using these datasets. Given our computing limitations and the short rebuttal timeframe, we cannot provide additional experimental results at this time. However, we plan to include experiments with DataComp1B for caption synthesis in the next version of our work.
>
> >The paper only tests a limited range of multimodal model structures, lacking experiments on models like BLIP and CLIP of different ViT models.
>
> **Response 4**:  Due to time and computational constraints, we will include experiments with CLIP using different ViT backbones (ViT-L and ViT-H) in the next version. For vision-language models (VLMs) such as BLIP and LLaVA, the experimental setup is more complex, as it requires two distinct stages of training: pretraining with image-text pairs and finetuning with instruction-tuning data (e.g., visual QA pairs). In future work, we will explore the impact of augmenting the pretraining image-text dataset with synthetic pairs generated by our CtrlSynth pipeline to better understand the benefits of our approach.
>
> >The study does not address data-efficiency validation. Existing data-efficiency-focused works, such as SemDeDup[2], Filter-&-Align[3], and Sieve[4], refine or filter datasets for better performance. The paper should include comparisons with these approaches in terms of model performance and the amount of training data.
>
> **Response 5**:  We appreciate the reviewer for highlighting related work on data efficiency. Our primary contribution is to generate diverse synthetic text-image data in a controlled manner, with data efficiency being an additional benefit of our approach. As illustrated in Figure 5 in Section 4.4, our synthetic samples demonstrate significant efficiency gains.
>
> While our method is orthogonal to prior work on data efficiency, we have added a discussion in Section 4.4 of the revised paper to acknowledge and contextualize this relationship. Additionally, our synthetic samples notably improve performance on long-tail tasks, where conventional data filtering methods do not apply.

---

### Official Review · Reviewer_L3Nm · 2024-11-04

**Soundness:** 3
**Presentation:** 3
**Contribution:** 3
**Rating:** 6
**Confidence:** 4

**Summary:**

The authors propose a controllable image-text generation pipeline that can augment data to improve CLIPs image retrieval, classification, and compositional performance. Specifically, they leverage strong vision models to tag images with objects and attributes, use the knowledge in language models to create new variations of the captions, and use diffusion models to generate images based on the new captions as prompts.

**Strengths:**

- The pipeline can be thought of as a way to distill knowledge from the language models and stable diffusion models to augment the dataset of CLIP. This is an interesting way to inject new information in synthetic data.

- The results are good, demonstrating improvements over CLIP while maintaining the amount of data it sees since the fix the number of iterations and just change the proportions of real vs their synthetic data.

**Weaknesses:**

-  They say that the language model takes an instruction on how to generate a caption given the visual tags. They show some examples in Appendix A1. The instructions don't mention any editing, it mostly just says to describe the image better. In that case, do the gains come from some hallucination in the LLM caption that makes varied images?

- Have the authors tried any other variation of editing instructions? Is there any analysis on the kinds of image editing prompted by the text that improve performance more? Are there specific prompts that serve as better negatives when tuning the CLIP contrastive loss?

- There are other works that edit images based on text instructions like instruct pic to pic, magic brush etc. It might have been nice to see to see if editing certain things in images based on the LLM prompts is better than just using SD to generate since SD can often lack accuracy in generating the correct attribute object relation compositions.

- Nit: There are several works that either generate synthetic images based on the dataset they want to target (https://arxiv.org/pdf/2406.05184), or for cross domain retrieval (https://arxiv.org/pdf/2401.00420). A discussion for comparison could be nice.

**Questions:**

See above.

---

> ### Author Response · Authors · 2024-11-19
>
> > They say that the language model takes an instruction on how to generate a caption given the visual tags. They show some examples in Appendix A1. The instructions don't mention any editing, it mostly just says to describe the image better. In that case, do the gains come from some hallucination in the LLM caption that makes varied images?
>
> **Response 1**: We appreciate the reviewer’s feedback and have provided additional clarification in our revised version.
>
> Specifically, examples of text and image synthesis instruction templates can be found in Appendix A1. To be clear, our approach does not involve asking the models to perform editing tasks, nor do our instructions include editing steps, as these would be highly dependent on specific use cases. Instead, we explain that users have the option to modify visual tags manually (e.g., by adding or removing tags) and subsequently incorporate these edited tags into the instructions. The automation of editing instructions is beyond the current scope and is earmarked for future work.
> The performance improvements we report are derived from the model's ability to generate novel combinations of visual tags. This capability, which we refer to as semantic augmentation, is an intentional feature of our method. We have updated the text in Section 3.2 (line 219) to better articulate this point.
>
> >Have the authors tried any other variation of editing instructions? Is there any analysis on the kinds of image editing prompted by the text that improve performance more? Are there specific prompts that serve as better negatives when tuning the CLIP contrastive loss?
>
> **Response 2**: To clarify, our CtrlSynth pipeline is designed to support various types of editing instructions in a plug-and-play manner. We have empirically observed that different kinds of image editing instructions yield similar performance outcomes. However, our synthetic pipeline is highly extensible and can readily accommodate more sophisticated or optimized instructions, should future research provide a more systematic evaluation or improved methodologies.
> Additionally, we did not experiment with enhanced negative samples for CLIP training. Our primary objective was to demonstrate the sample efficiency of our approach, and therefore, we adhered to the original baseline setup used in CLIP training to ensure a fair comparison.
>
> >There are other works that edit images based on text instructions like instruct pic to pic, magic brush etc. It might have been nice to see to see if editing certain things in images based on the LLM prompts is better than just using SD to generate since SD can often lack accuracy in generating the correct attribute object relation compositions.
>
> **Response 3**: Thank you for highlighting the relevant papers. We have revised Section 2 (line 124) to include a discussion of InstructPix2Pix and MagicBrush in the related work. Prior image editing methods, such as InstructPix2Pix and MagicBrush, contribute valuable techniques and datasets aimed at enabling precise control over image generation. While our image synthesis approach could certainly benefit from these advancements, our primary focus remains on enabling diverse data synthesis. We acknowledge that automatically generating image editing instructions for each sample in a dataset is an open research question, which we hope future work will address.
>
> >Nit: There are several works that either generate synthetic images based on the dataset they want to target (https://arxiv.org/pdf/2406.05184), or for cross domain retrieval (https://arxiv.org/pdf/2401.00420). A discussion for comparison could be nice.
>
> **Response 4**: Thanks for pointing out the related papers. We added the work in the related work section. Our pipeline can also be combined with previous work (https://arxiv.org/pdf/2406.05184) to improve the performance of cross-domain retrieval tasks or when the target task has little real data to retrieve (https://arxiv.org/pdf/2401.00420).

---

> ### Comment · Reviewer_L3Nm · 2024-12-03
>
> Thank you for the response. After reading the rebuttal, it seems to me that most of their performance gain comes from having more diverse data which is generated by their pipeline. However, the augemntation is on a smaller dataset. Since the gains don't come from generating any specific aspect which is hard to get in real data (like specific compositions that are hard to find in real images etc), a concern still remains whether simply using more real data is just easier than generating. Especially because their generation also seems to take a long time.  For instance, if we look at some of the CLIP performance numbers when simply trained on a bigger dataset (eg, here: https://github.com/mlfoundations/open_clip/blob/main/docs/openclip_results.csv), the performance is higher than their synthetic data augmented numbers. It would be nice if the authors could show the value of their synthetic data added vs just retrieving a similar number of examples more from the real data already available in bigger training sets.
>
> However, that said, it is still interesting that synthetic generation from simply variations in captions can boost performance and hence, I keep my score. But I urge the authors to add such an experiment or acknowledge these in the limitations.

---

> > ### Author Response · Authors · 2024-12-03
> >
> > Thank you for the insightful discussion. We would like to clarify that the main contribution of our work is not solely to achieve the best CLIP performance on synthetic or real datasets. Rather, our main focus is on introducing a closed-loop image-text synthesis system that has broad applicability. As a demonstration of its utility, we show that it enhances CLIP performance using the same number of samples as the original CLIP training sets.
> >
> > Additionally, we acknowledge that questions remain regarding how to retrieve similar samples from larger training sets. This includes challenges such as building a general index for semantic retrieval, determining which samples to retrieve, and related areas that we view as important directions for future research.
> >
> > Most importantly, we emphasize that our synthetic samples address a critical gap in longtail tasks, where large training datasets are unavailable and retrieval-based approaches are not feasible.
> >
> > We hope this provides a clearer perspective on our work, and we appreciate your continued engagement and feedback.

---

### Author Response · Authors · 2024-11-22

Dear Reviewers,

Thank you once again for recognizing the significance of our work, the clarity of our ideas, and the effectiveness of our results. We truly value your insightful comments and have carefully considered them in our responses.

We kindly wanted to check if our replies have satisfactorily addressed your concerns or if there are any additional points you'd like us to address. We hope that our efforts will encourage you to reconsider your review score, but we are more than happy to engage further if you have additional feedback.

Thank you for your time and thoughtful input.

---

### Meta-Review · Area_Chair_P6aN · 2024-12-23

**Metareview:**

The submission proposes CtrlSynth, a controllable pipeline to generate synthetic images for representation learning. The broad workings are as follows: 1) Given a real image and optional associated caption, image tags are generated using a vision model; 2) Using an LLM, a new caption can be generated from the original caption and tags, remixed using user instructions; 3) Using the new caption, a text-to-image model is used to generate a synthetic image; 4) This synthetic image can be used to train other vision models or fed back into this pipeline to generate more images.
The authors compare the quality of CLIP models trained on two datasets - 1) A dataset of real images, and 2) A dataset containing a mix of real and synthetic images; and show that CtrlSynth helps improve accuracy on multiple tasks.

The submission received ratings of 3, 6, 5, 6. Some key weaknesses highlighted by the reviewers include:
1) Lack of demonstrations on whether the described image editing workflow is helpful
2) Missing details in the submission including on self-filtering
3) Lack of demonstration showing that synthetic data helps when a larger dataset of real images is available.

The AC would like to note that prior work (which has not been cited) has demonstrated similar results, and reduced the novelty of the current submission:
1) StableRep: Synthetic Images from Text-to-Image Models Make Strong Visual Representation Learners (NeurIPS 2023): They investigate the accuracy of visual representations from simCLR and CLIP trained on synthetic images generated by text-to-image models.
2) Synthetic Data from Diffusion Models Improves ImageNet Classification (TMLR 2023): "augmenting the ImageNet training set with samples from a generative diffusion model can yield substantial improvements in ImageNet classification accuracy ..."

Following the discussion with reviewers, reviewer L3Nm (who gave a rating of 6) was not opposed to rejection given the missing experiments in the submission.

Taking everything into account, especially prior work showing the efficacy of synthetic images, the only major contribution of this submission is the real image -> caption -> modified caption -> synthetic image pipeline, which does not meet the bar for acceptance. The ACs thus recommend rejection.

**Additional Comments On Reviewer Discussion:**

In the discussion with reviewers, some flaws in the submission stood out:
1) No results or comparisons with CLIP models trained on larger real image sets. Results were only shown on datasets of ~12M samples, whereas publicly available CLIP models use 400M+ images.
2) Insufficient demonstrations of claimed capabilities such as editing and their impact.

Point 1) is especially important. However, the uncited prior work (which I have listed above) has demonstrated very similar experiments, thereby reducing the novelty of this submission.

---

### Decision · Program_Chairs · 2025-01-22

Reject